

# Hierarchical organization of a Sardinian sand dune plant community

Valentina Cusseddu[1], Giulia Ceccherelli[1] and Mark Bertness[2]

[1] Department of Science for Nature and Environmental Resources, University of Sassari, Sassari, Italy

[2] Department of Ecology and Evolutionary Biology, Brown University, Providence, Rhode Island, United States

## ABSTRACT

Coastal sand dunes have attracted the attention of plant ecologists for over a century, but they have largely relied on correlations to explain dune plant community organization. We examined long-standing hypotheses experimentally that sand binding, inter-specific interactions, abiotic factors and seedling recruitment are drivers of sand dune plant community structure in Sardinia, Italy. Removing foundation species from the fore-, middle- and back-dune habitats over three years led to erosion and habitat loss on the fore-dune and limited plant recovery that increased with dune elevation. Reciprocal species removals in all zones suggested that inter-specific competition is common, but that dominance is transient, particularly due to sand burial disturbance in the middle-dune. A fully factorial 2-year manipulation of water, nutrient availability and substrate stability revealed no significant proximate response to these physical factors in any dune zone. In the fore- and middle-dune, plant seeds are trapped under adult plants during seed germination, and seedling survivorship and growth generally increase with dune height in spite of increased herbivory in the back-dune. Sand and seed erosion leads to limited seed recruitment on the fore-dune while high summer temperatures and preemption of space lead to competitive dominance of woody plants in the back-dune. Our results suggest that Sardinian sand dune plant communities are organized hierarchically, structured by sand binding foundation species on the fore-dune, sand burial in the middle-dune and increasingly successful seedling recruitment, growth and competitive dominance in the back-dune.

## INTRODUCTION

Understanding the mechanisms biotic and abiotic that generate spatial patterns in natural communities is a major goal of ecology and is critical for developing ecology into a predictive science that can inform ecosystem management and contribute to conservation (*Morin, 2011*). Many natural communities are structured and defined by foundation species, sensu *Dayton (1975)*. Foundation species are defined operationally as common, abundant species that build and maintain habitats, ameliorating potentially limiting physical and biological factors, thus providing habitat for other species (*Jones, Lawton &*

Corresponding author
Valentina Cusseddu,
vcusseddu@uniss.it

*Shachak, 1994*; *Bruno & Bertness, 2001*; *Ellison et al., 2005*; *Angelini et al., 2011*). Examples of foundation species-dependent ecosystems include forests, coral reefs, salt marshes, mangroves, mussel and oyster reefs, which are all built and maintained by numerically-dominant habitat forming foundation species (*Bruno & Bertness, 2001*). Amelioration of potentially limiting physical and/or biotic conditions is a hallmark of foundation species-based ecosystems. Coastal sand dunes are physically-harsh habitats for the xerophytic plants that dominate them and are best understood as foundation species-based ecosystems (*Olff, Huisman & Van Tooren, 1993*).

Recently, it has been suggested that foundation species-based ecosystems are commonly hierarchical, where the amelioration of potentially limiting stresses is responsible for ecosystem establishment and maintenance, but that other species interactions are often responsible for generating the most conspicuous, but superficial spatial patterns in these communities (*Bruno & Bertness, 2001*; *Bruno, Stachowicz & Bertness, 2003*; *Altieri, Silliman & Bertness, 2007*; *Angelini et al., 2011*). While this model of community organization appears to be widespread (*Bruno & Bertness, 2001*), most evidence for hierarchical organization is anecdotal or correlative with few explicit experimental tests (for exceptions see *Altieri, Silliman & Bertness (2007)* and *Angelini & Silliman (2014)*). This is the case in spite of the potential importance of hierarchical community organization to conservation and management strategies. Plant communities that occur at the land/sea interface, like sand dune, salt marsh, sea grass and mangrove communities, provide important ecological functions including stabilizing shorelines from erosion and storm damage, harboring animal diversity, providing nursery habitats to threatened avifauna, marine turtles, and shellfish, as well as processing nutrient-rich terrestrial runoff (*Barbier et al., 2013*). Preserving these services belongs to informed management facing increasing threats. Sand dunes, however, are generally not conservation priorities and not managed to protect their socio-economic benefits (*Ehrenfeld, 1990*; *Everard, Jones & Watts, 2010*).

Historically most research on sand dune communities has been descriptive and reliant on dated correlative literature (e.g. *Cowles, 1899*; *Oosting & Billings, 1942*; *Mack & Harper, 1977*). Experimental work has been restricted to small-scale sand burial, seed dispersal and disturbance studies (*Maun & Perumal, 1999*; *Franks & Peterson, 2003*; *Miller, Gornish & Buckley, 2010*). Similarly, field studies of annuals and nurse plant effects, comparisons among chronosequences in plant species interactions (*Lichter, 2000*; *Franks, 2003*; *Cushman, Waller & Hoak, 2010*), experimental grazer studies (*Huntzinger, Karban & Cushman, 2008*) and plant/mycorrhizal associations studies (*Gemma, Koske & Carreiro, 1989*) have not examined large-scale patterns. Experimental studies of the roles of the dominant foundation species that have long been hypothesized to build and maintain sand dune plant communities by binding sand and ameliorating potentially limiting physical conditions such as water and nutrient limitation are notably absent. Consequently, the critical interactions between biological and physical processes that have been assumed to generate the organization of sand dune communities have not been rigorously tested.

Descriptive and correlative studies of sand dune communities suggest that substrate stabilization, water and nutrient limitation, and plant facilitation and competition are the

main drivers of sand dune community structure and organization (*Barbour, de Jong & Pavlik, 1985*; *Ehrenfeld, 1990*; *Lichter, 1998*; *Isermann, 2011*). Distinct plant zonation occurs in coastal sand dune systems (*Hesp, 1991*; *Lortie & Cushman, 2007*; *Acosta, Carranza & Izzi, 2009*). On the seaward border of sand dunes, the fore-dune, a limited number of clonal pioneer plant species with deep roots trap and bind sand, initiating dune formation. These plants stabilize substrate, trap seeds, and grow vertically and horizontally as sand accumulates, building the seaward border of sand dunes (*Cowles, 1899*; *Oosting & Billings, 1942*). At higher elevations in the middle-dune, plant species diversity increases, but unvegetated free space remains common. At these elevations, substrate stabilization remains important, but inter-specific plant interactions, including facilitation (*Franks, 2003*; *Castanho et al., 2015*) and competition (*Lichter, 2000*), and water and nutrient limitation also appear to mediate plant success. In the back-dune, furthest from ocean winds and salt spray, the dominant physical stresses of substrate instability and low soil nutrients and moisture are less severe, plant cover typically reaches 100%, and numerically-dominant woody shrubs or trees, appear to competitively displace plants that dominate lower dune elevations (*Lichter, 2000*). This descriptive structure of dune communities is consistent with a hierarchical community organization model and the stress gradient hypothesis of community assembly.

The hierarchical model of community organization (*Bruno & Bertness, 2001*; *Ellison et al., 2005*; *Altieri, Silliman & Bertness, 2007*) hypothesizes that within communities built by foundation species, dependent species are only able to persist through positive interactions and feedbacks initiated by the primary foundation species. The stress gradient hypothesis (*Bertness & Callaway, 1994*; *Maestre et al., 2009*; *He & Bertness, 2014*) proposes that the biological processes controlling community development shift from positive, facilitative interactions in physically and biologically harsh environments to negative, competitive interactions in benign environments. Many shoreline communities built and maintained by foundation species, such as salt marshes, coral reefs, mangrove forests, mussel reefs, and seagrass beds, have been hypothesized to be organized hierarchically by the general principles of the stress gradient hypothesis (*Bruno & Bertness, 2001*).

In this paper we examine long-standing assumptions of sand dune plant community ecology. We hypothesize that sand dunes are organized hierarchically, initially built by facilitation, but ultimately structured by spatially- and temporally predictable shifts from facilitative interactions to competitive interactions and seedling recruitment across decreasing stress gradients. Many of these patterns were initially theorized in classic descriptive work (*Cowles, 1899*), but have never been tested experimentally. We take a field experimental approach to test these underlying assumptions and elucidate the sand dune community assembly by examining three hypotheses that: (1) foundation species are responsible for stabilizing the seaward border of the dune from erosion and habitat loss; (2) at intermediate dune elevations, inter-specific plant interactions, plant resource availability, and seedling recruitment dictate plant abundance and distribution; and (3) at high dune elevations, reduced physical stresses lead to increased plant abundance, inter-specific competitive dominance and displacement (Table 1).

**Table 1** Objectives and corresponding tasks performed to develop our hypotheses.

| Objective | Task |
| --- | --- |
| **Site description** | • Plant zonation<br>• Plant elevation |
| **Hypothesis 1**: Hierarchical organization | • Foundation species removal experiment<br>• Temperature |
| **Hypothesis 2**: Stress gradient hypothesis | • Reciprocal species removal experiment<br>• Recruitment experiments (Seeds distribution, Seedlings survivorship, Seed transplant experiments) |
| **Hypothesis 3**: Competitive dominance | • Boardwalk shadow effect sampling<br>• Physical stress alleviation experiment<br>• Back dune competitive release experiment |

## MATERIALS AND METHODS

Our study was carried out on the Badesi dunes (40°56′45.571″N, 8°49′41.048″E) on the North coast of Sardinia, in the Mediterranean Sea. It is a wide dune system at the mouth of Coghinas River and has a plant community similar to other dune systems in Sardinia (V. Cusseddu, 2012, personal observations), it is within a Site of Community Importance (SIC) and the Town of Badesi gave us the field permit to undertake the field experiments (approval number: 3343 (23/03/2012)). The Coghinas dunes are almost 3 km long overlooking the Asinara Gulf, with a width of approximately 500 m. Sardinia has a warm temperate Mediterranean climate with hot, dry summers and cooler, wet winters (*Fadda, 2016*). As in other Mediterranean habitats, most plant growth and reproduction occurs during the winter months (*Blondel & Aronson, 1999*). Previous studies of Sardinian sand dune plant communities reveal that they have high species richness, endemism (*Bacchetta et al., 2008*; *Prisco, Acosta & Ercole, 2012*) and species adapted to physical stress (*Fenu et al., 2013*).

### Site description

To quantify the plant distribution across the Badesi dune, we surveyed transects at defined distance from the coast-line. According to our findings, we defined three major dune plant zones: the fore-dune, located on the seaward border of the dune and characterized by a steep lower border of stabilized sand and low plant diversity; the middle-dune, characterized by high but patchy plant species diversity and 40–60% bare sand cover; and the back-dune, characterized by total woody plant cover (Fig. 1). In addition to these zones we will also refer to the front of the fore-dune, or the seaward border of the fore-dune where embryo-dunes (*Cowles, 1899*) form and transition areas between major vegetation zones. To develop a site description of the plant community we estimated the change in species cover (plant zonation) and species elevation (plant elevation) across zones.

The plant zonation was assessed by quantifying the vegetation along 120 m, walking parallel to the coast-line, in each of four major dune zones: (1) the embryonic fore-dune,
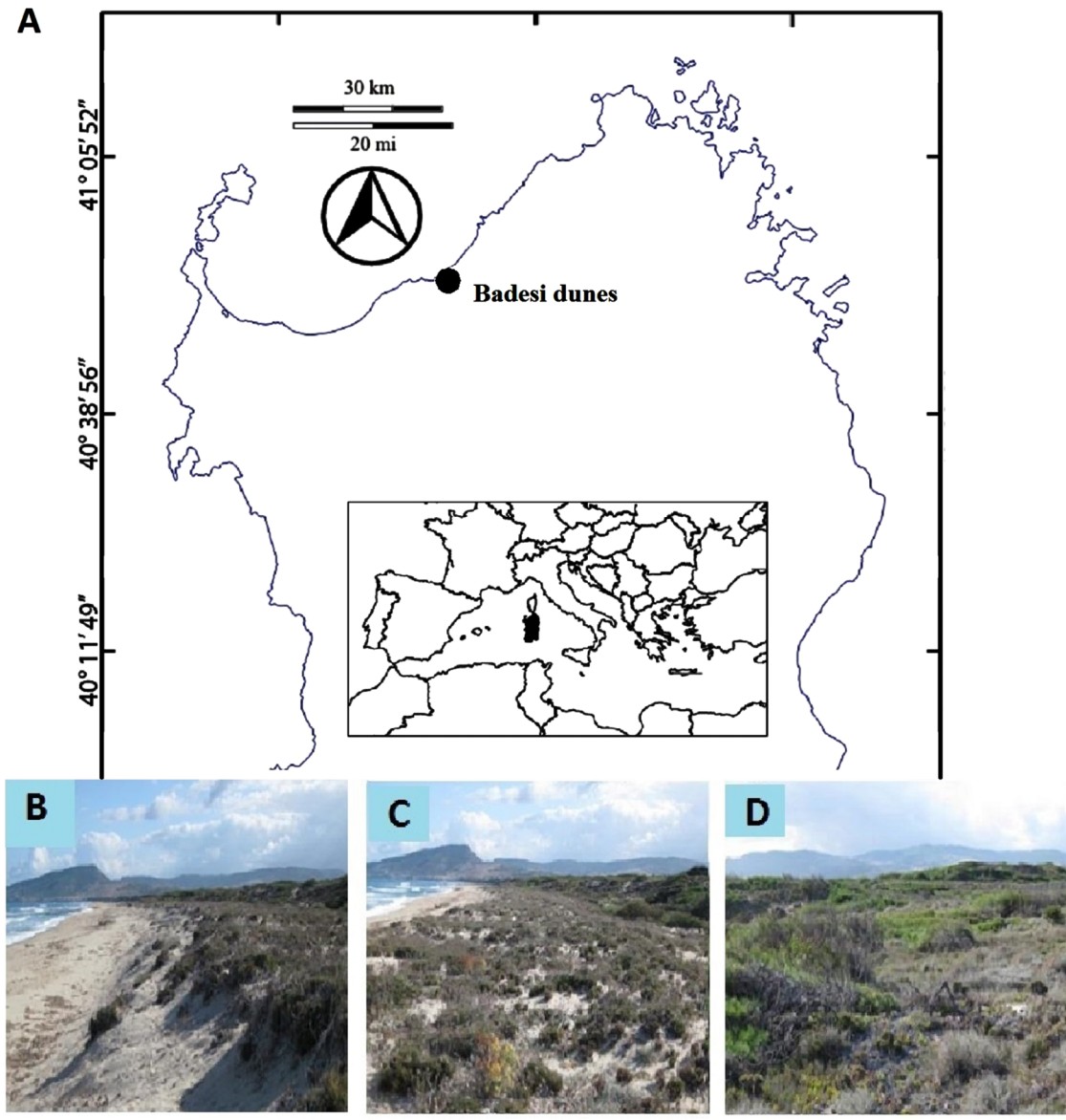

**Figure 1 Study site.** (A) Map of the location of the study site, and photographs of the dune zones at Badesi (Sardinia, Italy). (B) Fore-dune, (C) middle-dune and (D) back-dune.

(2) the top fore-dune on the first ridge of the dune formation, (3) the middle-dune with high plant species richness and bare substrate, and (4) the back-dune dominated by woody plants. In each zone, we ran a transect line, starting from a random point, parallel to the shoreline and every 2 m placed a $0.5 \times 0.5$ m sampling quadrat subdivided into a 25 $5 \times 5$ cm sampling grid to estimate plant species cover and unvegetated substrate cover. Sixty quadrats were sampled per zone. We characterized species diversity for each zone with the Shannon diversity index (H). All plant species, after first introduction, will be referred to by genus names.

Plant elevation has been estimated by a correlative approach to support the conspicuous feature of plants on the fore- and middle-dunes that generally occur on

elevated sand mounds. Since sand in these zones is mobile and windblown, we hypothesized that these mounds were created by the presence of plants rather than plant establishment on transient dune features (*Cowles, 1899*). We used an auto level and stadia rod to quantify the elevation of plants in each zone (n = 20/zone) and bare sand adjacent to (25-cm away) each plant. In each zone we selected 20 adult individuals of the most common plant species randomly, measured their elevation and the elevation of unvegetated sand 10–20 cm away. Plant and adjacent sand elevation differences were calculated, tested for homogeneity of variances with Cochran's test, transformed as necessary (*Underwood, 1997*) and analyzed by species with a one-way ANOVA, using the statistical package GMAV (*Underwood & Chapman, 1998*), and by a zone × plant ANOVA for *Armeria pungens*, the only species with an inconsistent effect across zones.

## Foundation species removal experiment

We performed a foundation species removal experiment to test the hypothesis that foundation dune plants bind sand and build the dune habitat, as well as to quantify secondary succession. In each of the three major dune zones in March 2012, we marked the corners of 24 randomly placed 1 × 1 m plots each separated by at least 10 m. Within each zone, each plot was individually labeled and then randomly assigned to one of three treatments: (1) controls, (2) total species removal, or (3) procedural controls (n = 8/treatment/zone). The four corners of each plot were marked with 2 cm diameter rebar driven to refusal into the sand and cut to initially extend 10–15 cm above the sand surface. Control plots were otherwise untouched. In plant removal replicates, plots were perimeter trenched to 30 cm with straight edged shovels and all vegetation in the plots was sprayed weekly for three weeks with a systemic herbicide (Roundup, Monsanto) until all vegetation was dead. Dead above-ground vegetation was left to simulate natural death, so to evaluate the consequences of the lack of activity in terms of erosion. Roundup is widely used in ecological research, has localized effects if plots are trenched to avoid translocation outside of the target area, and we have used it successfully in the past in shoreline habitats (*Bertness & Hacker, 1994*). Procedural controls were trenched, but not herbicided.

Surface Elevation Table (SET) posts (*Cahoon et al., 2000*) modified for experimental replication were installed in the center of all plots to measure sand erosion/accretion as a function of the presence/absence of foundation species and dune zone. A 2 cm diameter rebar rod was driven to refusal in the center of each plot, cut 10–15 cm above the sand surface and fitted with a 30 cm long horizontal PVC bar with four evenly spaced locations to measure the height of the sediment (see *Brisson, Coverdale & Bertness (2014)*). Elevations were taken in October and March of each year at each SET post for three years. Corner post heights were simultaneously measured to supplement SET data and assess spatial patterns in sand erosion/accretion. Sediment height (the sand erosion/deposition balance) data were analyzed with a two-way ANOVA of zone and treatment, both considered as orthogonal fix factors, followed by post-hoc testing (SNK test).

In the spring of 2013, we began monitoring temperature after noting apparent summer heat death of some high middle-dune plants during the first year of this experiment.

We deployed 24 thermistors (Econorma S.a.s. FT-800/System) to plots of this first experiment (n = 8/dune zone), attaching them under the canopy in control plots and plant removal plots with wire staples. This allowed us to quantify plant heat exposure during the summer, and quantify differences between control and removal treatments. Thermistors were left in the field from the mid-June until the first week of September 2013 measuring temperature hourly. We took into account the data between 01:30 a.m. and 05:30 a.m. and between 13:30 p.m. and 17:30 p.m. to estimate the lowest and the highest daily temperature, respectively. The average of these temperature ranges gave us mean minimum (Min) and mean maximum temperatures (Max) for each zone and treatment. Temperature was analyzed with a three-way ANOVA considering zone, range (Min vs Max), and treatment (removal vs control) as orthogonal and fix. A one-way ANOVA was used to analyze mean maximum temperature by zone.

## Reciprocal species removal experiment (inter-specific plant interactions)

To test the hypothesis that inter-specific plant interactions shifted from facilitative to competitive across the sand dune, being especially important in the middle-dune, we performed reciprocal species removal experiments in all dune zones with dominant species pairs (see plant zonation). On the fore-dune, we chose *Armeria* (sea rose) and *Lotus cytisoides*; in the middle-dune we chose *Armeria*, *Lotus*, and *Carpobrotus acinaciformis* (ice plant); and in the back-dune, we chose *Armeria*, *Carpobrotus*, and *Pinus* spp. (*Pinus halepensis* and *Pinus pinea*). For each species pair in each zone, we located 24 0.5 × 0.5 m plots with mixtures of the two target species. All plots were marked with numbered rebar corner posts driven to refusal into the sand and labeled with a unique numbered plastic tag. For each zone and species pair (species a and b), we randomly assigned control plots, "species a" removal plots, and "species b" removal plots (n = 8/treatment/species pair/dune zone). Species assigned for removal were pulled, manually when possible, with minimal disturbance. Treatments were maintained monthly as needed for two years. During this time, plots were photographed in the spring and fall of each year and analyzed for percent plant cover. The height of the corner posts was also measured to quantify sand deposition/erosion. Plant cover was analyzed by zone with a two-way ANOVA, only *Carpobrotus* data were Sqrt (X + 1) transformed, to meet the assumption of homoscedasticity. Separate ANOVAs were run for each species at 12 and 18 months, to accommodate the loss of plots to sand burial over time. For the same reason, we also analyzed plant cover and sand deposition on plot plant cover data pooled by zone with a t test and plant × sediment cover with linear regression.

## Seedling recruitment

In October 2013, sand under plants and in adjacent bare sand > 25 cm from plants, in all dune zones, have been sampled to examine the distribution of dune plant seeds and test the hypothesis that like sand, seeds are deposited and trapped under adult plants. This was done after the summer when seed dispersal and germination was most pronounced on the dune (*Bákker, Bravo & Mouissie, 2008*). In each zone (fore, middle and back) we

sampled sand under and adjacent to the most common plants in each zone by taking 100 ml surface sand samples (1 cm deep; n = 10/species/habitat/zone). Samples were returned to the laboratory and sorted under a dissecting microscope. Seed density was analyzed with an experimental treatment (under and adjacent to adult plants) one-way ANOVA separately by zone and on conspecific seeds of the target plant and seeds of all other plants combined.

Seedling survivorship across the dune has been examined in order to test the hypothesis that seedling survivorship increased with dune elevation and association with adult plants, we marked natural seedlings in all dune zones that were under adult plants and in bare sand > 30 cm from an adult plant. A total of 450 seedlings were marked and monitored monthly for survivorship over three seasons. Seedling survivorship was analyzed with a non-parametric log rank test to compare survival among zones (fore vs middle vs back) and species survival by location (next to adult vs adjacent bare sand). In both cases the whole follow up period was taken into account (*Bland & Altman, 2004*) and a $\chi^2$ test was done on the log rank data of Ln (X + 1) transformed *Armeria* seedling data.

A seed transplant experiment with the five most common Badesi sand dune plants (*Armeria*, *Lotus*, *Carpobrotus*, *Cakile maritima*, and *Pancratium maritimum*) was performed to examine the hypothesis that seed supply is a determinant of the distribution and abundance of plants across the dune. For each species, we collected dehiscing seed heads and dissected out and separated the seeds of each species. We then sorted them into aliquots of 4–10 seeds depending on species seed availability and placed them by species into polyester mesh bags that would retain the seeds, but would allow germination and seedling growth. For each species we planted 21 seed bags in each zone (fore, middle and back) under conspecific adults and in unvegetated sand > 30 cm from adult vegetation and marked their location with color-coded wire markers. We monitored seed transplants weekly for germination and seedling survivorship for three months. Germination and seedling survivorship were analyzed separately by species with non-parametric log rank data, using a $\chi^2$ test.

We also transplanted seeds of *Pancratium* in mesh bags (n = 10), loose in the sand (n = 10), and loose in the sand covered on the sand surface with nylon mesh (2 mm mesh, 5 × 5 cm cover pinned to the sand surface with garden fabric staples; n = 10) to test the hypothesis that loose seeds were eroded away on the fore-dune. We only did this experiment in the fore-dune since the fore-dune was the only zone that showed significant erosion. Seed species were marked and identified by color-coded wires (2 mm). All seeds were planted 3 cm below the sand surface. We monitored these transplants for germination and survivorship weekly for the first three months and monthly for one year. Germination and seedling survivorship data were analyzed with a non-parametric log rank data, using a $\chi^2$ test.

**Boardwalk shadow effect sampling**

During the first year we noticed that in the summer months, temperatures in the middle and back-dune, protected from on shore winds by the fore-dune berm, were extremely high and coincided with the death of the *Carpobrotus* at high dune elevations. To examine

the hypothesis that shading would enhance the dominance of *Carpobrotus*, we quantified the long-term effect of shading on the Badesi dune as a proxy for high temperature impacts. In February 2015, we quantified shaded and unshaded vegetation adjacent to boardwalks (1.5 m wide, elevated 50 cm over the substrate) perpendicular to the shoreline that extended from the fore-dune to the highest point of the back-dune, and 4 m on both sides of the boardwalks. We hypothesized that shading by the boardwalk would decrease solar stress on *Carpobrotus*. We sampled two boardwalks that had been in place five years. At each boardwalk we quantified live and dead *Carpobrotus* % cover in $0.5 \times 0.5$ m quadrats every 2 m from the beginning of the fore-dune to the back-dune. For analysis we excluded plots levels without *Carpobrotus* and pooled the data from adjacent and control plots. Dead/alive ratio of *Carpobrotus* % cover and dead *Carpobrotus* % cover were Sqrt $(X + 1)$ transformed to meet the assumptions of parametric statistics and analyzed with a one-way ANOVA.

## Physical stress alleviation experiment

The hypothesis that physical stress limits dune plant recovery across the dune was tested by running a fully factorial experiment across all zones manipulating all combinations of nutrient limitation (with 33 ml of slow release Osmocote NPK pellets spread on them every six months and without nutrient additions), water limitation (with 2 liters of tap water every 2–3 weeks and without water additions) and substrate stability (with and without substrate stabilizing fish net attached flush to the surface with wire staples). In each zone we located 72 $0.5 \times 0.5$ m plots with bare sand substrate. Every combination of nutrient, water, substrate limitations and controls were marked and labeled with numbered rebar corner posts (n = 8/treatment combination/zone). Replicate treatments were maintained for two years and all plots were monitored for % plant cover initially and then for two years in the spring and fall. Plant cover in the plots was Ln $(X + 1)$ transformed to meet parametric statistic assumptions and analyzed with a treatment × dune zone ANOVA.

## Back-dune competitive release experiment

It was not possible to do *Pinus* removal experiments similar to the reciprocal plant species removal experiments or foundation species removals in other zones because of the size of *Pinus* trees. Since the Badesi dune is a protected conservation area, removing entire *Pinus* trees would have been destructive and not permitted. A competitive release experiment was conducted, to test the hypothesis that *Pinus* domination of the terrestrial high dune border is due to competitive dominance, by removing large (~2 m long) *Pinus* branches and estimated natural plant recruitment by following seed germination and cover in $1.5 \times 1.5$ m plots under *Pinus* canopies (n = 10), where *Pinus* shading was alleviated by branch removal (n = 10) and in areas where *Pinus* shading was removed but replaced by a similar level of shading by shade cloth (n = 10), as procedural controls. A central $0.5 \times 0.5$ m quadrat in each plot was monitored photographically monthly for a year.

We also manipulated sediment in the *Pinus* plots described above, to test the hypothesis that allellopathy contributes to the dominance of *Pinus* and *Carpobrotus* in the back-dune.

Allelopathy was suggested because in dense *Pinus* and *Carpobrotus* stands, natural substrate is covered by *Pinus* needles and *Carpobrotus* leaves, but seedlings of all species are extremely rare (G. Ceccherelli, 2013, personal observation). In 100 ml plastic greenhouse seedling pots we planted seeds of *Armeria*, *Pancratium*, and *Lotus* (plus no seed controls) with either (1) *Pinus* soil, (2) middle-dune bare soil, (3) middle-dune *Carpobrotus* soil, (4) potting soil mixed with sand, (5) potting soil with a 2 cm layer of *Pinus* needles and (6) potting soil with a 2 cm layer of *Carpobrotus* leaves (n = 10/*Pinus* treatment/soil type) and scored them monthly for germination. Seed germination and survivorship of *Armeria*, and *Lotus* were transformed with Ln (X + 1) and all were analyzed by ANOVA testing the interactions between *Pinus* treatment × soil type.

## RESULTS

### Site description

Elevational zonation of plants across the Badesi dune is striking (Fig. 2). The fore-dune has low plant richness and cover (35%) and 65% unvegetated sand cover. *Armeria* (sea rose), *Otanthus maritimus* (cotton weed) and *Lotus* (trefoil of the cliffs) are the numerically dominant plant species on the fore-dune and all are clonally spreading, deep-rooted perennials. The width of the fore-dune varies at Badesi from ∼20–25 m. The middle-dune has over 28% higher plant cover and 53% higher species richness than the fore-dune, but still has considerable bare sand substrate (Fig. 1, 38%). *Armeria*, *Carpobrotus* and *Ephedra distachya* (joint pine) are the most common middle-dune plant species and are all clonally spreading perennials. The middle-dune is ∼30–35 m wide.

The transition from the middle to back-dune is more gradual (Fig. 2). The seaward border of the back-dune has the highest plant species richness on the dune, and is dominated by the ice plant, *Carpobrotus*, a perennial succulent, that can be seen overgrowing other back-dune plants like *Pinus* on the terrestrial border of the dune (33%). At higher elevations of the back-dune, *Pinus* dominates the landscape as a solitary evergreen species that has a prostrate morphology at lower elevations, an arborescent morphology at higher elevations and an understory of bare substrate in the dense *Pinus* canopy that dominates high elevations.

Substrate topography in relation to plants varied across the dune. In dune zones closest to the water, plants were found on elevated sand mounds that decreased in elevation with distance from the water, while in the back-dune, furthest from the water, plants were not associated with sand mounds. On the front dune, *Armeria* and *Otanthus* were found on sand mounds $53 \pm 2.16$ cm (mean $\pm$ SE) and $56 \pm 2.51$ cm higher than adjacent substrate, respectively. For the front dune, the one-way ANOVA did not reveal any significant difference between the two species ($F_{1,38} = 0.36$, $p > 0.05$). On the fore-dune, *Lotus* and *Armeria* were similarly found on mounds $17 \pm 4.64$ and $24 \pm 1.92$ cm high, respectively, and no significant effect was detected with the ANOVA ($F_{1,38} = 2.07$, $p > 0.05$). In contrast, in the middle-dune, *Armeria*, *Carpobrotus*, *Lotus*, *Helicrysum microphyllum* and *Ephedra* were found on smaller sand mounds $22 \pm 1.47$, $12 \pm 2.06$, $11 \pm 1.06$, $16 \pm 1.27$, and $12 \pm 2.48$ cm high, respectively. In this zone the difference among species was significant ($F_{4,95} = 7.51$, $p < 0.0001$, SNK: *Armeria* > all others).

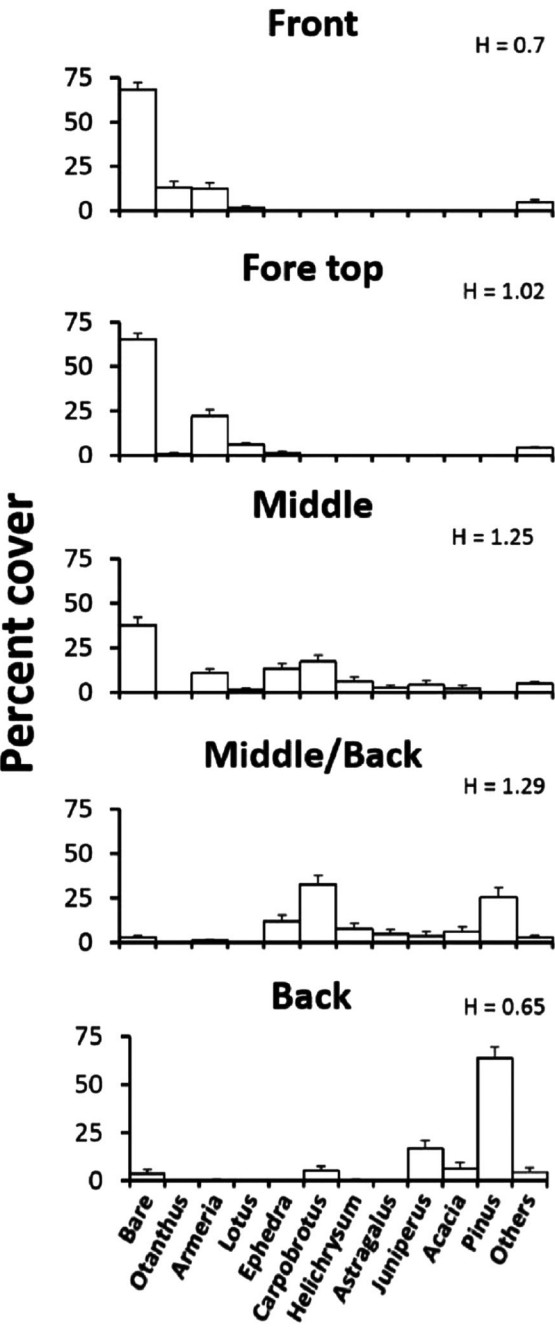

**Figure 2 Plant zonation.** Plant zonation (mean percent cover + SE) at the Badesi dune. Transects parallel to the shoreline were sampled at 2 m intervals with 0.5 × 0.5 m quadrats (60/zone) to estimate percent cover. H is the Shannon diversity index.

*Armeria*, a robust perennial shrub, was found on higher sand mounds than all other species. One-way ANOVA on *Armeria* revealed differential *Armeria* sand binding by zone ($F_{3,76} = 111.36$, $p < 0.0001$, SNK: front > fore top = middle > back). In the back-dune, buffered from winds and sand transport by the lower dune zones, *Armeria* = 9 ± 1.65 cm, *Carpobrotus* = 4 ± 1.31 cm, *Pinus* = 4 ± 2.76 cm, *Acacia cyanophylla* = 0 ± 3.8 cm, and *Juniperus* spp. (*Juniperus macrocarpa* and *Juniperus*

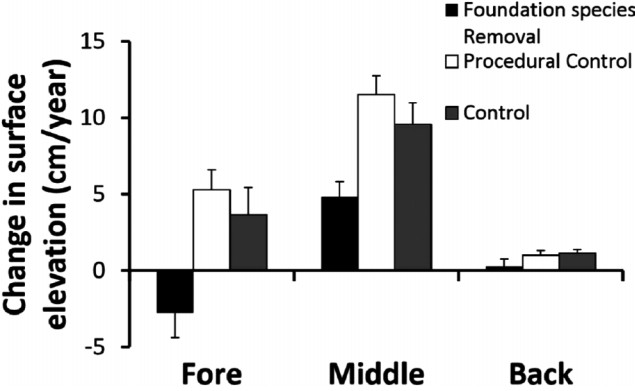

**Figure 3 Foundation species removal experiment.** Surface elevation or sand erosion/deposition balance (mean + SE) of the foundation plant species removal experiment in the three dune zones, measured for three years.

phoenicea) = −2 ± 4.72 cm were not found on elevated locations and there were no significant species effects ($F_{4,95}$ = 1.93, p > 0.05).

## Foundation species removal experiment

Removing foundation species had different effects across zones (Fig. 3). The two way ANOVA revealed a significant zone × treatment interaction ($F_{4,63}$ = 2.67, p < 0.05). In the fore-dune, foundation species removal led to sand erosion of > 2 cm/year, in contrast to control and procedural control plots that had annual sand accretion rates of > 4 cm/year (Fig. 3; p < 0.05, SNK test fore: removal < procedural control = control). Sand erosion in the fore-dune foundation species removal plots was dramatic and led to the collapse of the fore-dune edge in just two years. When foundation species were removed, erosion on the seaward edge corner posts increased over 60% (compared to procedural control and control plots) leading to an amount of sand dispersion of −2.8 ± 1.6 cm/year (Fig. 3).

In the middle-dune, sediment level variation resulted in net sand accretion that occurred in all treatments (Fig. 3; p < 0.05, SNK test: fore = back < middle), but was nearly twice as high in control and procedural control plots with live vegetation to bind and trap sand than in removal plots (SNK middle: removal < procedural control = control).

In the back-dune, annual sediment accretion was more than an order of magnitude less than middle-dune plots with foundation species (Fig. 3). Sand accretion in the back-dune was also similar among foundation removal, control and procedural control treatments (Fig. 3; p < 0.05, SNK back: removal = procedural control = control). In all treatments, annual accretion was < 1 cm/year and when foundation species were removed there was no sand accretion.

Mean minimum temperature (Min) did not differ among treatments or zones (fore 18.73 ± 0.45 and 18.50 ± 0.4 °C, middle 19.11 ± 0.28 and 19.12 ± 0.18 °C, back 16.59 ± 0.77 and 17.07 ± 0.87 °C for removal and control, respectively; $F_{2,228}$ = 1.77, p > 0.05 for the interaction zone × temperature range × treatment). In contrast, mean maximum temperature (Max) differed among zones (fore 33.73 ± 1.40 and 32.83 ± 2.46 °C, middle 41.86 ± 2.74 and 38.31 ± 1.34 °C, back 44.89 ± 3.61 and 40.26 ± 3.45 °C

for removal and control, respectively; $F_{2,114} = 39.85$, $p < 0.0001$, SNK: fore < middle = back) peaking in back zone removal treatments ($F_{1,114} = 10.72$, $p < 0.002$, SNK: Removal > Control). Maximum temperatures peaked around 70 °C, and averaged 45 °C in the back-dune removal plots, 5 °C higher than when vegetation was present.

## Reciprocal species removal experiments (inter-specific plant interactions)

Reciprocal species removal experiments initially revealed significant inter-specific interactions (Fig. 4). Over time, however, these interactions were lost due to sand burial and erosion disturbance, particularly in the fore and middle-dune zones as evidenced by a decrease in plant cover over time in the species interaction plots (Fig. 4; middle t = −3.84 and $p < 0.001$, t test). After 18 months, 22% of the middle-dune reciprocal species interaction plots had been completely lost due to sand burial, and an additional 10% of the plots were still recognizable but almost totally buried, while no plots were lost in the fore or back-dune. To analyze this experiment, we ran separate two-way ANOVAs for each species after 12 and 18 months of treatments, taking into account species interaction (comparison of each pair in which the species is involved) and treatment (reciprocal species removal vs control) as orthogonal and fixed factors. Insignificant interaction p values will always refer to the last sampling date (18 months). In the fore-dune after 6, 12, and 18 months, there was no evidence of reciprocal effects between *Armeria* and *Lotus* (Fig. 4; interaction × treatment $F_{4,70} = 1.96$, $p > 0.05$ and $F_{2,42} = 2.27$, $p > 0.05$, respectively).

In the middle-dune, after one year of treatment, *Carpobrotus* cover increased in both pairs where *Armeria* and *Lotus* were removed, but after two seasons this effect was no longer apparent (Fig. 4; treatment $F_{3,56} = 12.12$, $p < 0.03$, SNK reciprocal species removal > control at 12 months, interaction × treatment $F_{3,56} = 1.23$, $p > 0.05$ at 18 months). At the same time, there was no evidence of interactions affecting *Armeria* or *Lotus* (Fig. 4; interaction × treatment $F_{4,70} = 1.96$, $p > 0.05$ and $F_{2,42} = 2.27$, $p > 0.05$, respectively). In all the inter-specific interaction plots in the middle-dune there was, over time, a general decrease in plant cover associated with sand burial disturbance (Fig. 4).

In the back-dune sand burial was not prevalent, but a summer die off of *Carpobrotus* was seen every year (see below). After 12 months *Armeria* cover decreased in absence of both *Carpobrotus* and *Pinus* but at 18 months this pattern was no longer apparent (Fig. 4; interaction × treatment $F_{4,70} = 3.31$, $p < 0.02$, SNK test for both Back pairs: reciprocal species removal < control at 12 months, $F_{4,70} = 1.96$, $p > 0.05$ at 18 months). While, after one year of removal, the opposite was found for *Carpobrotus*, whose cover has increased in the absence of both *Armeria* and *Pinus*, but after another two seasons of observation this growth was no longer detectable (Fig. 4; treatment $F_{3,56} = 12.12$, $p < 0.03$, SNK reciprocal species removal > control at 12 months, interaction × treatment $F_{3,56} = 1.23$, $p > 0.05$ at 18 months). Reciprocal removal of *Pinus* in the back-dune did not reveal significant species interactions (Fig. 4; interaction × treatment $F_{1,28} = 0.18$, $p > 0.05$).

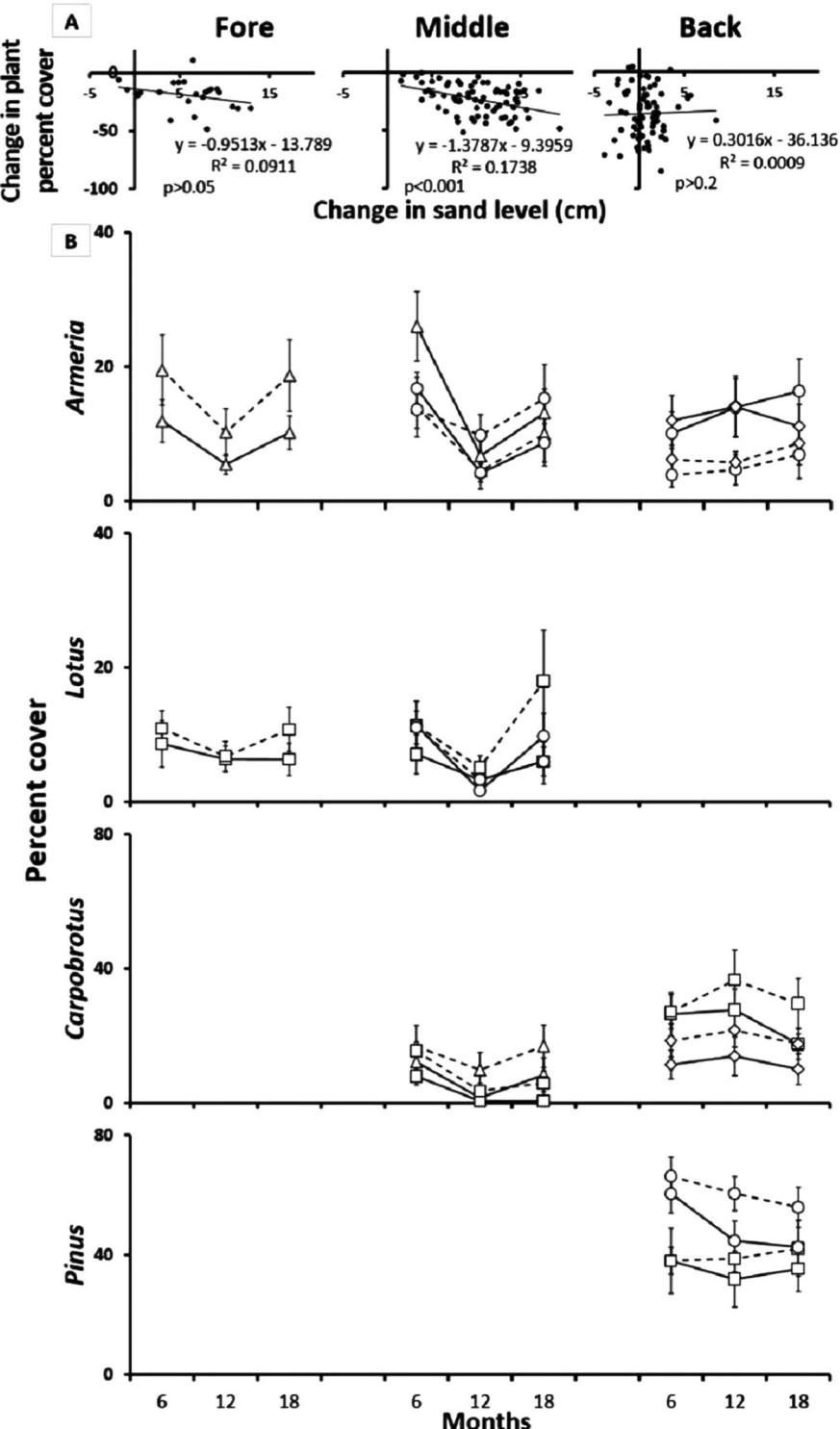

**Figure 4 Reciprocal species removal experiments.** (A) Relationship between sand burial and percent plant cover in the pair-wise reciprocal species removal experiments after 12 months in the fore, middle and back dune. (B) Reciprocal species removal: *Armeria pungens* (□), *Lotus cytisoides* ( △ ), *Carpobrotus acinaciformis* (○), *Pinus* spp. (◇), Species control (—), Reciprocal species removal (- - -). Results of pair-wise reciprocal species removal experiments after 18 months in the fore, middle and back-dune.

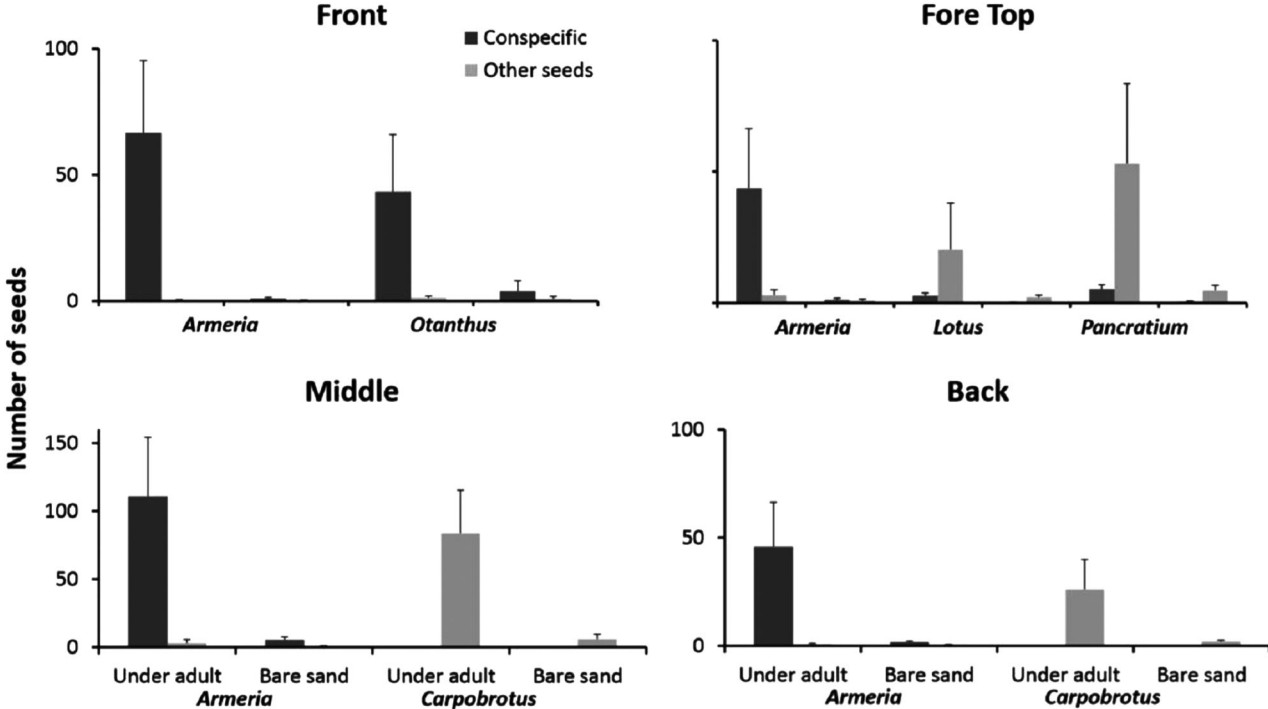

**Figure 5  Seeds distribution experiment.** Seeds distribution (mean + SE) of common dune plants across zones under adult plants and in adjacent unvegetated sand. Data are given separately for conspecific and non-conspecific seeds.

## Seedling recruitment: seed and seedling distribution

Seeds of the most common dune plants were most abundant under adult plants independent of species identity and dune zone (Fig. 5). For each zone, we performed a three-way ANOVA considering the species, the substrate (under adult vs bare sand) and the seeds category (conspecific vs other seeds) all as orthogonal and fixed factors. We will present only the factors and/or the interactions that were significant (complete results at Table S1).

On the front of the fore-dune, seeds of *Armeria* and *Otanthus* were almost exclusively found under conspecific adults, whereas the seeds of other species were rare and not found associated with adult plants (Fig. 5; substrate × seeds interaction $F_{1,72} = 6.56$, $p < 0.02$). On the top of the fore-dune, seeds of *Armeria* were more common under *Armeria* adults than in nearby bare sand, but for *Lotus* and *Pancratium*, seeds of conspecifics were just as common in bare sand than under adults (Fig. 5). For *Lotus* and *Pancratium*, conspecific seeds were equally distributed under and away from adults, but seeds of other species were significantly more common under *Lotus* and *Pancratium* than in adjacent bare sand (Fig. 5; species × seed category interaction, $F_{2,108} = 4.23$, $p < 0.02$ and species × substrate × seeds category $F_{2,108} = 3.45$, $p < 0.04$, respectively).

In the middle-dune, *Armeria* seeds were more common under *Armeria* adults than in adjacent bare sand, but seeds of other species were not (Fig. 5), while for *Carpobrotus*, conspecific seeds were rare, but seeds of other species were more than an order of magnitude more abundant under *Carpobrotus* than in bare sand

(Fig. 5; species × substrate × seeds interaction, $F_{1,72} = 12.59$, p = 0.0007). In the back-dune, *Armeria* seeds were 20 times more common under conspecifics than in adjacent bare sand, while for *Carpobrotus*, seeds of other species were significantly more abundant under *Carpobrotus* than in adjacent bare substrate (Fig. 5; species × substrate × seeds interaction $F_{1,72} = 8.34$, p < 0.006).

The survivorship of marked seedlings was analyzed with a log rank test comparing the survival of the seedlings of each species comparing the proximity with adult (next to adult vs adjacent bare sand) and of the seedlings in general pooled in three groups corresponding to the three zones (fore vs middle vs back seedlings), on the times (weeks) of survival events. The percent of seedlings survivorship increased dramatically with dune zone elevation but was not affected by neighboring plants (Table S2; $p < 0.05$, $\chi^2$). In the fore-dune, no *Lotus* seedlings survived (independent of the proximity to adult neighbors), no *Pancratium* seedlings survived on bare sand, and less than 5% of *Pancratium* seedlings survived next to adults, but this result was not significant ($p > 0.5$, $\chi^2$). In the middle zone only 5% of *Lotus* seedlings survived with or without adult neighbors ($p > 0.5$, $\chi^2$), while for *Armeria* 25% of marked seedlings in bare sand survived, nearly three times (10%) the number of seedlings that survived next to adults ($p < 0.025$, $\chi^2$). No *Pancratium* seedlings survived. In the back-dune survivorship of *Armeria* and *Carpobrotus* seedlings was over 80%, far higher than any other zone, and was not influenced by the presence or absence of neighbors (Table S2).

## Seedling recruitment: seed transplant experiments

Of the six dune plant species seeds used in transplant studies, three species, *Pancratium*, *Lotus*, and *Cakile* germinated. Only 2 and 3 *Armeria* and *Otanthus* germinated from back and fore-dune planted seeds, respectively, while *Astragalus thermensis* and *Carpobrotus* did not germinate in any location or treatment. For each of the three species that had sufficient germination, we compared the germination and survival in each of the three zones. About 30–40% of *Pancratium* seed transplants germinated, but germination was similar among zones ($p > 0.1$, $\chi^2$). Survivorship of seedlings from the seed transplant experiment showed that all species had the highest survivorship in the back-dune, but this pattern was only significant for *Pancratium*, which had the highest sample size (Fig. 6; $p < 0.005$, $\chi^2$). Less than 10% of the transplanted *Lotus* seeds germinated and *Lotus*, found ubiquitously across the dune, had higher germination in the back-dune, and lowest germination in the middle-dune (Fig. 6; $p < 0.025$, $\chi^2$), while survivorship did not differ among zones ($p > 0.5$, $\chi^2$). For *Cakile*, a pioneer species found naturally in the fore and middle-dune, 12–18% of transplanted seeds germinated and among zones had higher germination in the middle than in fore and back-dune (Fig. 6; $p < 0.05$, $\chi^2$). Survivorship, however, did not differ among zones ($p > 0.5$, $\chi^2$).

In the fore-dune seed stabilization experiment with *Pancratium*, germination and survivorship were analyzed comparing germination and survival among the three treatments (seeds in bags vs loose seeds vs loose seeds with net covers). Due to erosion, germination was highest in the bagged seed treatment, second highest in the seed treatment with stabilized substrate (net cover), and lowest in the loose seed treatments

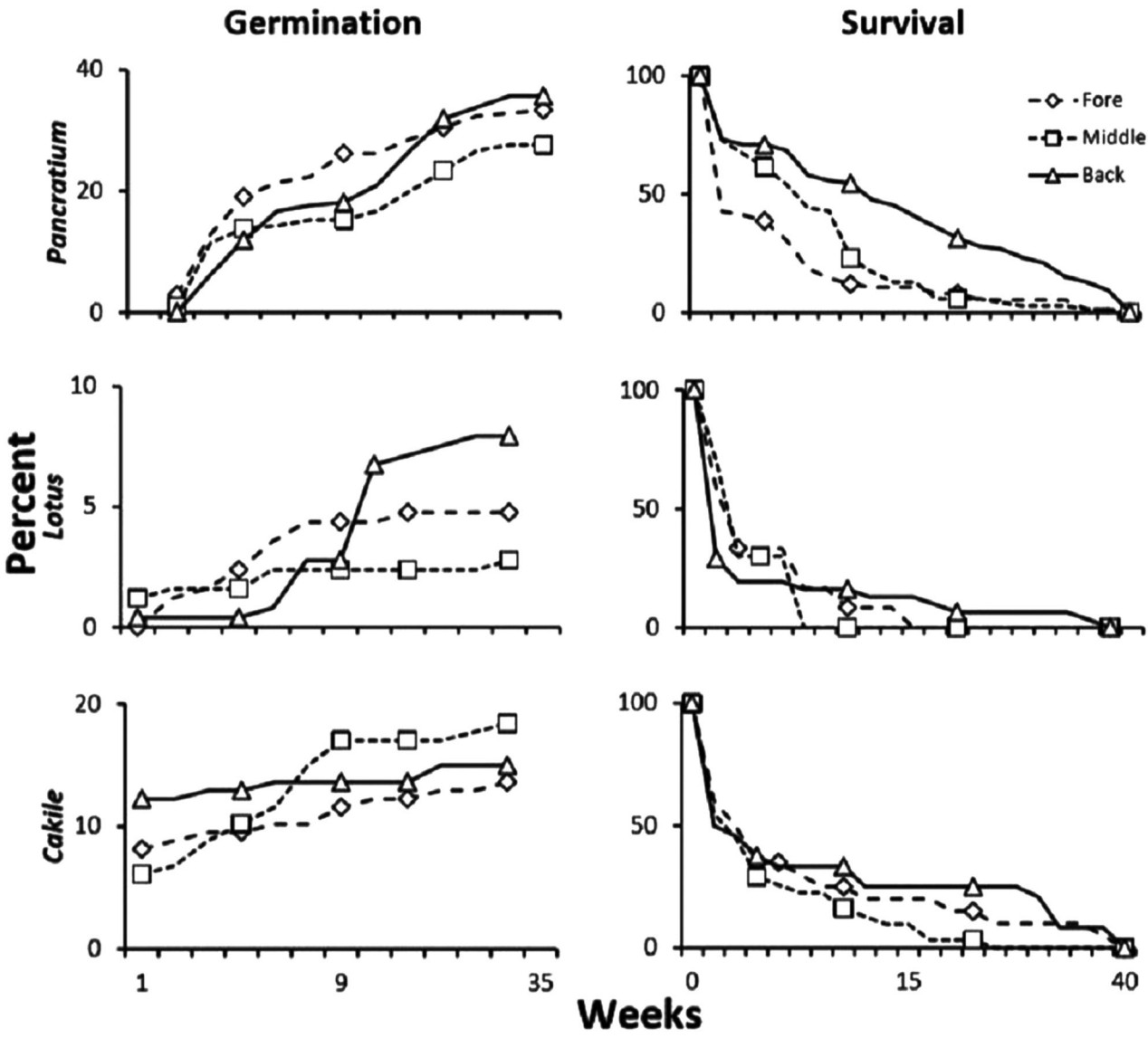

**Figure 6 Seed transplant experiment.** Results of the seed transplant experiment where germination permeable net bags of locally collected seeds (n = 21/zone/species) from dehiscing seed heads were transplanted to the fore, middle and back-dune with and without plant neighbors and scored for germination and seedling survivorship. Data are percent survivorship and germination of the total potential.

(Fig. S1; p < 0.005, $\chi^2$). The difference in germination between loose seeds and the secured seed treatments was > 10%, suggesting that sand erosion in the fore-dune can lead to seed loss. Seedling survivorship in this experiment was similar among stabilization treatments (p > 0.25, $\chi^2$).

**Boardwalk shadow effect sampling**

Analysis of *Carpobrotus* cover data (dead/alive ratio) adjacent to and 2 m away from the shade of boardwalks revealed almost twice as much live *Carpobrotus* cover under the shade of the boardwalks (46 ± 3.24%) than in adjacent unshaded habitats (25 ± 2.84%, $F_{1,190}$ = 5.58, p < 0.02). There was also nearly 10% more dead *Carpobrotus* in unshaded habitats (16 ± 1.76%) than under the shade of boardwalks (7 ± 1.10%, $F_{1,190}$ = 18.76, p < 0.0001).

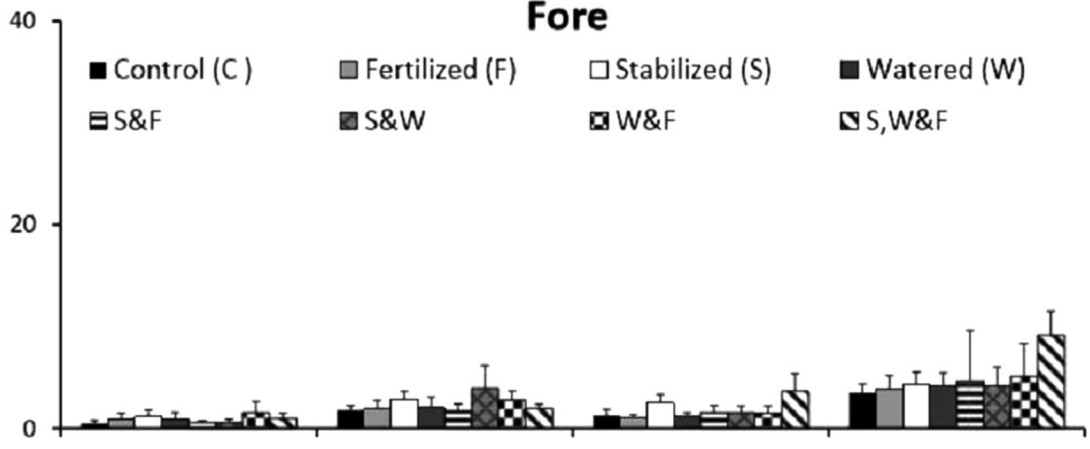

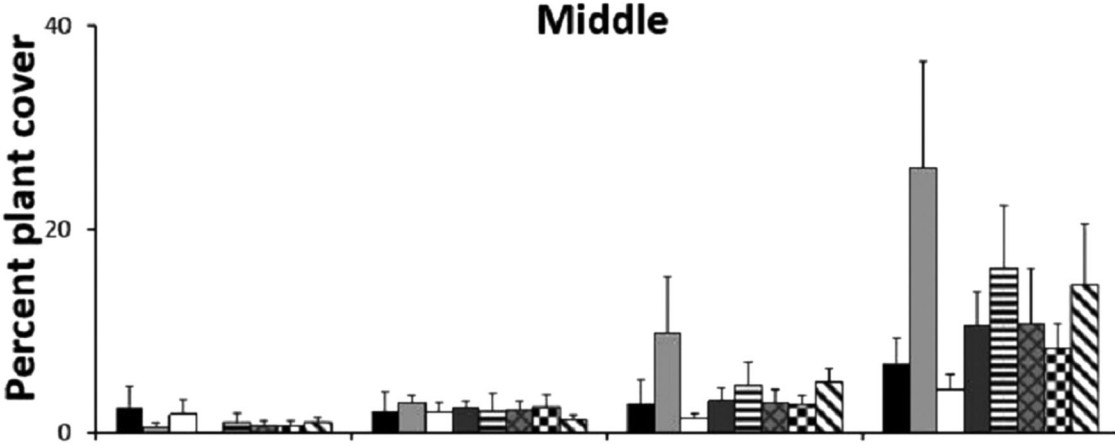

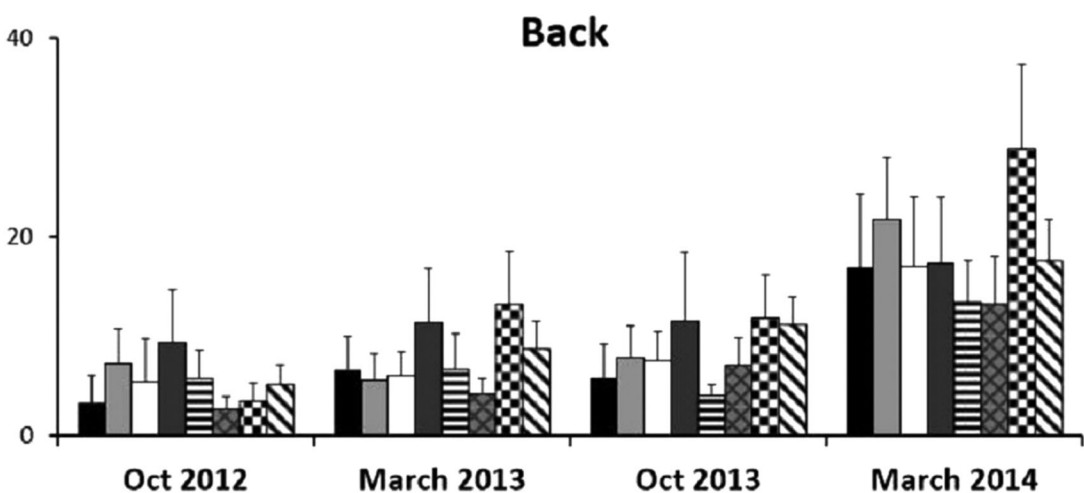

**Figure 7 Physical stress alleviation experiment.** Percent plant cover (mean + SE) of the fully factorial manipulation of water availability, nutrient availability and substrate stability during two years in the fore-, middle- and back-dune. No single or combination of stress alleviation treatments affected plant cover, but higher recovery occurred in back zone.

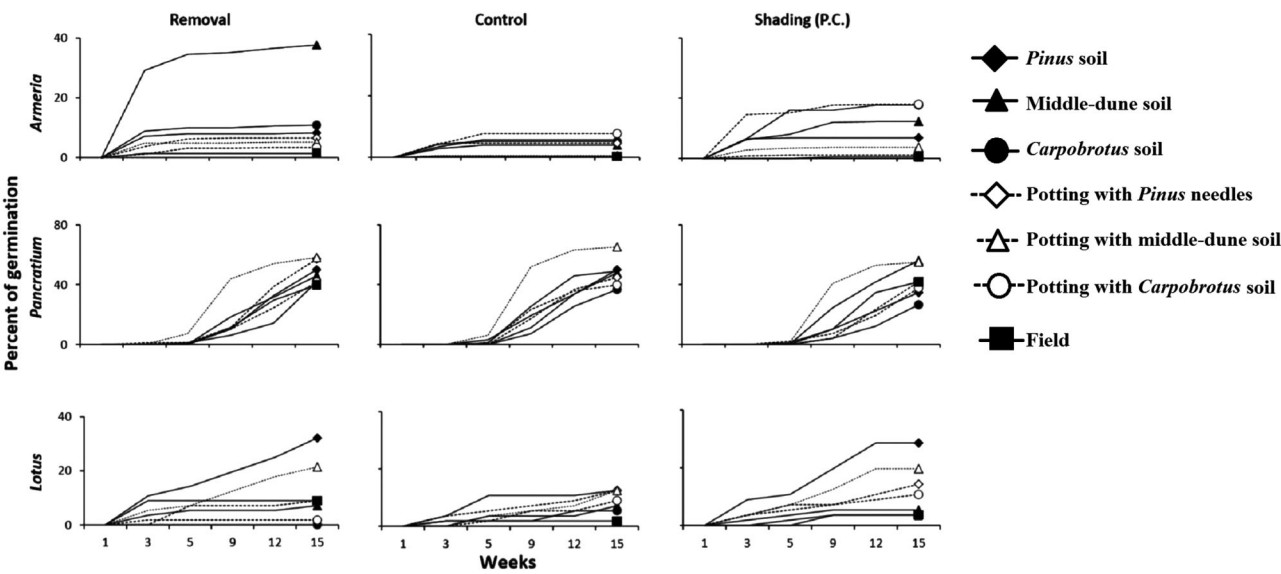

**Figure 8 Back dune competitive release experiment.** Germination of *Armeria*, *Pancratium* and *Lotus* seeds sowed in different soils and in Treatment field (no canopy), Control (under canopy) and Procedural controls (*Pinus* removal with shade cloth to simulate canopy shading, P.C.). N = 8/treatment. Data are percent of seeds germinated out of the total potential.

## Physical stress alleviation experiment

Experimentally manipulating water availability, nutrient availability and substrate stability across dune zones (fore, middle and back) in a fully factorial two year experiment at a 0.5 × 0.5 m spatial scale revealed that these factors, long thought to be critical to sand dune plant communities, had no effect on plant recovery, as evidenced by a two-way ANOVA (Table S3). However, recovery increased from the fore to the back-dune (Fig. 7; Table S3, SNK fore < mid < back).

## Back-dune competitive release experiment

Natural plant recruitment was not observed under *Pinus*, irrespectively of canopy removal treatments. However, germination of experimentally planted seeds under *Pinus* occurred, although it was generally lower in control plots, especially for *Armeria* and *Lotus* (Fig. 8). We ran a two-way ANOVA by species with canopy treatment and soil type as fixed and orthogonal factors. Under the *Pinus* canopy, *Armeria* seeds attracted small animals as they were often removed or eaten. *Armeria* germination, however, was significantly higher in the *Pinus* removal treatment in middle-dune soil, and very little germination occurred in any soil type under *Pinus* canopy and in procedural control plots (Fig. 8; canopy treatment × soil type interaction, $F_{12,126} = 2.54$, $p < 0.005$). *Pancratium* and *Lotus* germination was not influenced by the *Pinus* treatment ($F_{2,126} = 2.11$, $p > 0.05$ and $F_{2,126} = 0.79$, $p > 0.05$, respectively), but they were significantly affected by soil type (Fig. 8; $F_{6,126} = 5.02$, $p = 0.0001$ and $F_{6,126} = 7.06$, $p < 0.0001$, respectively), even if, for both species, there were no significant post hocs comparisons. The interaction *Pinus* treatment × soil type was not significant for both species ($F_{12,126} = 0.96$, $p > 0.05$ and $F_{12,126} = 0.84$, $p > 0.05$, respectively).

## DISCUSSION

Our results, testing the basic assumptions of sand dune community organization, generally support a hierarchical model of sand dune community organization and confirm many, but not all of the assumptions of early descriptive and correlative studies on the organization of these communities. Experimental examination of community assemblages typically has not upheld the findings of earlier correlative conclusions. For example, rocky intertidal, salt marsh, kelp forest, and coral reef community dynamics once thought to be driven by physical forces (e.g. *Odum, 1953*; *Lewis, 1964*; *Mann, 1982*), were later shown to be regulated strongly by interactions between physical and biological factors once community level experiments were carried out (*Dayton, 1971*; *Estes & Palmisano, 1974*; *Bertness & Ellison, 1987*). Our results suggest that fore-dune foundation species are responsible for binding, stabilizing and maintaining sand dune communities, and that inter-specific plant interactions occur across the dune, but are generally overshadowed by physical stresses, particularly sand burial in the middle-dune. Additionally, they indicate that while physical stresses limit plant distributions across the dune, their impacts are largely stochastic, except in the back-dune where competitively dominant woody plants competitively displace other sand dune plants. In the back-dune species removal plots, the most conspicuous trend over time was the decrease in the cover of *Carpobrotus* during summer. This left large areas of dead, desiccated *Carpobrotus* dominating the seaward border of the back-dune at the end of the summer. Since *Carpobrotus* death occurs almost exclusively in summer (V. Cusseddu, 2013, personal observation), mortality at high dune elevations appears to be driven by heat and desiccation from direct sun exposure leaving the desiccated plants marking their initial distribution.

### Biogenic communities, foundation species and hierarchical organization

Like most communities in physically and/or biologically stressful habitats, the Badesi sand dune plant community is dependent on foundation species that ameliorate stress and facilitate community development (*Angelini et al., 2011*). The front, fore and middle-dune dominant plants occur on elevated sand mounds that appear to be actively trapped by passive deposition of wind-blown sand. Our foundation plant species removals reveal that plant species on the fore-dune bind and stabilize sand, building the habitat that supports all the dune plants living at higher elevations, making them foundation species-dependent (Fig. 3). Without habitat-modifying foundation species that initially colonize potential fore-dune habitats, sand dune plant communities would not develop or be maintained. Biogenic communities that are dependent on foundation species often display this type of hierarchical organization (*Bruno & Bertness, 2001*; *Angelini et al., 2011*) and include salt marshes (*Angelini et al., 2011*), terrestrial forests (*Ellison et al., 2005*), seagrass meadows (*Duarte et al., 2000*), as well as smaller scale associations within communities (*Angelini & Silliman, 2014*). This common dependency of communities on foundation species habitat modification needs to be incorporated into ecological theory (*Bruno, Stachowicz & Bertness, 2003*). It is a major organizing force in community

assembly, confirmed by the overwhelming evidence that it is the template for communities in physically and biologically stressful habitats due to habitat amelioration and associational defenses, respectively (*Bertness & Callaway, 1994*; *Ellison et al., 2005*; *Crotty & Bertness, 2015*).

## Seedling recruitment

Sexual recruitment is generally problematic in physically- and biologically-stressful communities like salt marshes (*Pennings & Callaway, 1996*) and coral reefs (*Hughes & Jackson, 1985*). As a result, asexual reproduction and clonal growth play a leading role in the community dynamics of such communities. Sexual recruitment is limited in Sardinian sand dune plant communities exposed to severe substrate mobility and strong wind and salt spray exposure. Dispersed seeds of dune plants were trapped near substrate stabilizing adult plants and were uncommon in unvegetated substrate (Fig. 5). The survivorship of marked seedlings was low, but generally increased from the fore to the back-dune and was higher when seedlings were associated with adult plants than when on bare sand substrate without neighbors.

Field germination experiments with three common species had low germination rates and poor survivorship in all zones except the back-dune. Seedlings of back-dune grew the best, but suffered from desiccation, due to high temperature and herbivory (Fig. 6). Like in other stressful environments, the low success of seed reproduction appears to lead to nurse plant effects (*Franks, 2003*) and strong reliance of dune plants on clonal reproduction, particularly at low elevations (*Maun, 1998*). Nurse plant effects have been identified as a strong generator of pattern in sand dune plant communities in general (*Martinez, 2003*).

## Competitive dominance

Pairwise quantification of the interaction of the numerically common plants in each dune zone revealed that inter-specific plant competition was common across the dune landscape. In the middle-dune, however, burial disturbance over the 18-months time course of our experiments revealed that, while inter-specific plant interactions occurred, their importance was transient and not of long-term consequence in generating species distribution patterns. Rather, sand burial disturbance limited the role of inter-specific competition and competitive displacement, but promoted plant species diversity and coexistence in the middle-dune (Fig. 4). Similar results have been found on rocky shores, intertidal boulder fields, mussel beds, salt marshes, and grasslands (see *Dayton, 1971*; *Sousa, 1979*; *Paine & Levin, 1981*; *Bertness & Ellison, 1987*; *Platt, 1975*, respectively). In the back-dune, where sand burial is less common, competitive dominance by woody plants emerges as a major factor leading to competitive dynamics that determine the prevalence of shrub and woody species like *Pinus* and *Armeria* and exclusion of other dune species. Due to the dense *Pinus* canopy, germination of other middle-dune species is precluded and in some cases soil type prevents seedling recruitment. Moreover, high summer temperatures in the back-dune limit *Carpobrotus* competitive dominance, because it suffers drying and summer die-off limiting its dominance, favoring *Pinus*.

## Sand burial disturbance

While our inter-specific plant interaction experiments initially revealed significant interactions (Fig. 4), sand burial, particularly in the middle-dune, was ultimately the most pervasive and powerful driver of plant community patterns at lower dune elevations, rendering competitive dominance and facilitative interactions inconsequential. Thus, the same physical force that is responsible for building the dune, sand mobility, is responsible for plant mortality and diversity, particularly in the middle-dune. Burial in sand is recognized as a major aspect shaping the arrangement and composition of vegetation in coastal sand dune communities (*Ranwell, 1958*; *van der Valk, 1974*; *Maun & Lapierre, 1986*). Sand deposition has been renowned as a main selective force in the evolution of seeds, in survivorship of seedlings and adult plants and, to a larger scale, in zonation and succession of vegetation (*Maun, 1994*; *Maun, 1998*).

## Lessons from Sardinian sand dunes

Our results reveal that sand dune plant communities are organized hierarchically and dependent on the establishment of fore-dune foundation plants that bind sand above and belowground, leading to sand dune initiation, development, and maintenance (Fig. 3). Without foundation plant species, mechanisms such as sand binding, erosion, mobility, and burial act as barriers to plant community development (Fig. 3) and are more pronounced than at higher elevations that are less exposed to sand mobility. On the fore-dune, removing the foundation species led to habitat loss, while competitive and facilitative plant species interactions were not detectable due to sand erosion and burial disturbance (Fig. 4). Manipulating physical factors thought to be critical in sand dune communities (i.e. water additions, nutrient additions, substrate stabilizations) in well replicated plots of all these factors alone and in combination also did not affect plant colonization at the small 0.25 m$^2$ spatial scale manipulated, this does not mean that, at other scales, would not have an effect, or that water and nutrient availability are unimportant for the structure of dune communities. Seed and seedling success were also rare on the fore-dune suggesting that asexual clonal expansion and colonization was more common in the highly disturbed fore-dune habitat. This also suggests that disturbance on the fore-dune trumps all other biological and physical factors. Since all of these physical and biological factors have been shown to correlatively impact sand dune communities at regional spatial scales, our results imply that spatial scale is important in understanding process and pattern in sand dune ecosystems since they are so fundamentally shaped by disturbance processes and their interaction with foundation plant species that stabilize the substrate.

Plant species cover and richness increased from 35% in the fore-dune to 63% in the middle-dune (Fig. 2). Like the fore-dune, however, recovery from foundation removal was minimal, being less than 1% in 3 years. Moreover, while the middle-dune initially revealed inter-specific plant interactions, it is subjected to heavy sand burial that limited the role of species interactions, seed germination (Fig. 6), and seedling success (Table S2). Therefore, frequent sand burial disturbances have a large influence in the middle-dune zone on the community dynamics (Fig. 4).

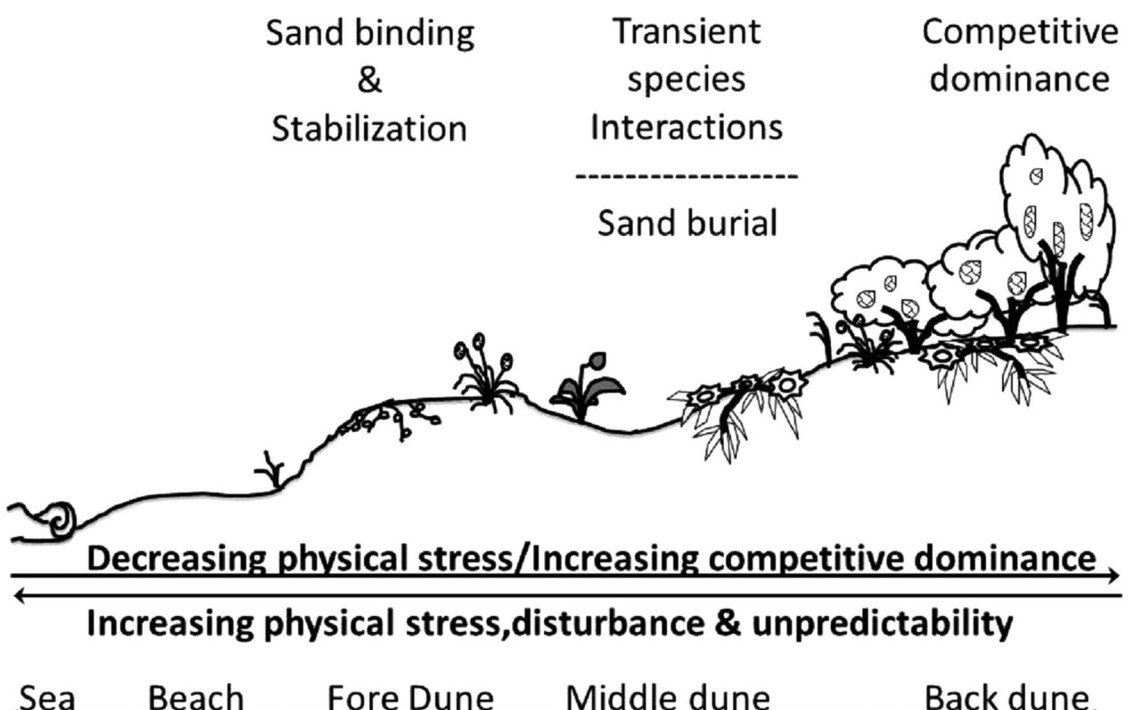

**Figure 9 Conceptual model of Badesi dunes.** Conceptual model of the assemble rules of the Badesi sand dune plant community.

In the back-dune we see nearly 100% plant cover and the emergence of competitive dominants (Figs. 1D and 2). On the seaward border of the high dune the *Carpobrotus* is common and appears to be competitively dominant to middle-dune plant species, but, at higher dune elevations, the evergreen *Pinus* competitively displaces most other dune plants, forming a back-dune *Pinus* monoculture. Seedling germination and survivorship were higher in the back-dune than at lower dune elevations (Fig. 6), as was the recovery of the species (Fig. 7). The competitive dominance of *Pinus* in the back-dune was shown by a competitive release experiment in which removing *Pinus* canopy in the back-dune lead to the success of important foundation species such as *Armeria*, especially favored in the middle-dune soil (Fig. 8). In general, however, seeds of plants characteristic of the middle and fore-dune planted in *Pinus* zone showed a quite good capability of germination. This suggests that they are excluded from the back-dune because *Pinus* acts as a physical barrier, having a dense prostrate morphology (Fig. 8).

## CONCLUSIONS

Our results are summarized in a conceptual model of the assembly rules of the Badesi sand dune plant community (Fig. 9). The gradient of stress in this dune system acts crosswise, from land to sea and back again, creating an area of aggregation of the various factors in the middle of the dune, in which the living conditions are particularly difficult. It is for this reason that facilitation mechanisms are determined among species, however, they are transient in nature due to high instability. Physical stress on the vascular

plants of the dune, including wind exposition, sand scouring, and other stressors of terrestrial origins, decreases with increasing dune elevation exposure. Conversely, the role of biological factors in shaping the dune plant community increases with increasing dune elevation. These factors include inter-specific plant competition and facilitation, increasing seedling recruitment and survival, and increasing herbivory. The Badesi dune plant community is a biogenic community built and maintained by foundation plant species that stabilize sand on the fore-dune. On the middle-dune, sand burial disturbance is the dominant structuring force that limits the role of sexual recruitment and inter-specific competition in structuring the middle-dune zone. In the back-dune, *Carpobrotus* and *Pinus* competitively dominate all other dune plants, but high summer temperatures limit *Carpobrotus* dominance in the back-dune, resulting in *Pinus* competitively dominating the back-dune terrestrial border.

Ecosystems, like sand dunes, where the establishment of foundation species ameliorates stresses allow community development that would not occur without foundation species. Coral reefs (*Hughes & Jackson, 1985*), salt marshes (*Bruno & Bertness, 2001*), mangroves (*Ellison & Farnsworth, 2001*), seagrass (*Duarte et al., 2000*), and forest ecosystems (*Ellison et al., 2005*) have all been explicitly or implicitly described as hierarchically organized systems. In all of these ecosystems, once foundation species enabled community establishment, while other biotic factors like competition, recruitment, and trophic dynamics became important community structuring forces (*Bruno & Bertness, 2001*). These findings and their implications are important for understanding community organization well enough to make it a more predictive science and for conservation since they stress that managing, protecting, and restoring foundation species should often be the first step in many conservation efforts, rather than focusing on charismatic species.

## ACKNOWLEDGEMENTS

We would like to thank H. Chen, S.M. Crotty, E. Farris, R.S. Filigheddu, E. Suglia, T. Pettengill, M. Bergland and S. Hagerty for discussion and comments, T.C. Coverdale, S. Pinna, S. Oliva, F. Bulleri, L. Piazzi, J. Bernardeau, J. Boada for assistance with field and laboratory, S. Pisanu for help identifying plants, S. Ramachandran for statistical advice and the mayor of Badesi, A.P. Stangoni, for his support. This work was part of V. Cusseddu's PhD.

### Funding

The Italian Ministry of Education and Research (MIUR), PRIN 2012–2015 (TETRIS), University of Sassari (Ateneo's PhD and Ulisse Fellowship), the Robert P. Brown Chair in Biology, Brown University, and a Fulbright Fellowship supported this work. The funders had no role in study design, data collection and analysis, decision to publish, or preparation of the manuscript.

## Grant Disclosures

The following grant information was disclosed by the authors:
Italian Ministry of Education and Research (MIUR).
PRIN 2012–2015 (TETRIS).
University of Sassari.

## Competing Interests

The authors declare that they have no competing interests.

## Author Contributions

- Valentina Cusseddu conceived and designed the experiments, performed the experiments, analyzed the data, wrote the paper, prepared figures and/or tables, reviewed drafts of the paper.
- Giulia Ceccherelli conceived and designed the experiments, performed the experiments, analyzed the data, contributed reagents/materials/analysis tools, reviewed drafts of the paper.
- Mark Bertness conceived and designed the experiments, performed the experiments, contributed reagents/materials/analysis tools, wrote the paper, reviewed drafts of the paper.

## Field Study Permissions

The following information was supplied relating to field study approvals (i.e., approving body and any reference numbers):
Town of Badesi approval number: 3343 (23/03/2012).

## Data Deposition

The raw data has been supplied as Supplemental Dataset Files.

## Supplemental Information

Supplemental information for this article can be found online at http://dx.doi.org/10.7717/peerj.2199#supplemental-information.

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
