# Peer review of "Hierarchical organization of a Sardinian sand dune plant community"

_PeerJ, doi:10.7717/peerj.2199_

## Round 0.1 · original submission · Minor Revisions

This paper was enjoyable to read and reports a very thorough study of the process structuring community assembly in a sand dune community. Frequently offered in textbooks a case study of the process of succession the authors are right to state there have been few experimental/manipulative studies of this kind. This is an important paper in that regard

The paper describes a fairly large number of related experiments that are gradually introduced through the methods section. I would like to see clear objectives for each of these provided in the Introduction. Given the complex nature of the paper, with inter-related hypotheses being tested, it would be clearest if a numbered, or bullet-point list of objectives were provided - one for each experiment.

The reviewers suggest that there is possibly too much in this paper and I agree that there definitely some sections that seem more relevant than others to the story they want to tell. Looking at the discussion section I think you should be able to tell where cuts can be made - doing so will make the overall message clearer whilst only sacrificing a number of "add-n" experiments.

Many of the Methods paragraphs start "To examine" or "To test" which gets a little wearying, can the language be made a little less repetitive?

As the reviewers suggest there is a clear need to improve the quality of the figures and it's questionable whether they're all needed. I did like the Figure 12 though and thought it gave a nice summary of the findings.

Other specific comments and suggestions can be found in the annotated paper attached

·

Basic reporting

The submission appears to adhere to all PeerJ policies.
The article is written in English using clear and unambiguous text and must conform to professional standards of courtesy and expression. But it could be improved.
The article includes sufficient introduction and background to demonstrate how the work fits into the broader field of knowledge. Relevant prior literature should be appropriately referenced.
I don't know the templates but this looks pretty good to me.
Figures are relevant to the content of the article, of sufficient resolution, and appropriately described and labeled. But could be improved
The submission ia ‘self-contained,’ represent san appropriate ‘unit of publication’, and includes all results relevant to the hypothesis. Coherent bodies of work should not be inappropriately subdivided merely to increase publication count.
All appropriate raw data has been made available in accordance with our Data Sharing policy.

Experimental design

The submission describes original primary research within the Scope of the journal.
The submission defines research questions, they are relevant and meaningful. The knowledge gap being investigated identified, and statements are made as to how the study contributes to filling that gap.
The investigation has been conducted rigorously and to a high technical standard.
Methods should be described with sufficient information to be reproducible by another investigator.
The research is conducted in conformity with the prevailing ethical standards in the field.

Validity of the findings

The paper is a weighty one and I suspect it could be pruned a little and this would probably make it more effective. But I like it overall. Good paper.

On a grammar note I am a total pedant and like lots of hyphens (shorten your word count!) and I don't like split infinitives - sorry I know I am probably the old one left!

I have made some suggestions to the text to help improve clarity. See attached sheet (hope you can read my writing).

I have some issues with the reproduction of the anovas, not because I think they are wrong but they are not clear to me from the text.

You said you were going to refer to plant species by genus - but then use ice-plant- I would stick to your original suggestion. Carpobrotus


L. 71 sentence isn't clear to me
L.74-83 Long sentence - suggest rewording
L108 what is biogenic?
L103 Leave this to the discussion.
L211 I don't follow your sentence - what data are being analysed in this three way anova i.e. what is y in y~a*b*c
L367-369 this is or should be in methods
L445-455 You start analysing this as a oneway (?) but then move to a 2-way Surely you start with the two way and if the interaction is significant this is the contrasts of interest. The one-way responses are irrelevant - or have I misunderstood. If so reword for clarity.

Indeed it might be one way to shorten the manuscript is to pout in a table of anova tests for each set of measurements analysed. The you could just state Anova was used to test for treatment effects (Table 2). It could even be in a supplementary appendix. I really did get confused with the way you have written it. I am not suggesting you have done it wrong, just find it hard to follow. You could also in the Table identify fixed and orthogonal effects. This would reduce wordage in the text and helpimprove flow.

You also don't specify the package you used to do the analyses in the Methods.

The graphs need to be improved.
Fig. 1-3 OK
Fig. 4, 5 could you not put them together on one figure. You discuss them together and it is irritating to move between them suggest you put the three panels on Fig 5 along the top of fig 4.Perhaps reduce the size of the symbols in Fig.5.
Fig. 6-7 OK
Fig. 8 Might be better as a Table - Data are percent presumably refers to the numbers but this is unclear as the graphs themselves represent data.
Fig 9. Suggest you reduce the symbols to a much smaller size as they obscure the trends.
Fig. 10 +/- SEs would be a good addition if possible
Fig. 11 again I would reduce the size of the symbols as they make the lines look unclear Or just use the lines with perhaps one symbol at end of line. It is messy and will difficult to read in print.
Fig. 12 This could be improved a lot to improve visual impact.

Additional comments

Interesting paper, looks well done. comments above are to help improve the final draft. Minor corrections I don't want to see it again.

Reviewer 2 ·

Basic reporting

• The language is clear and professional!
• Introduction and background is okay. However, see specific comments.
• Literature: I am not sure I fully understand the difference between Grime’s old but relevant theory of stress, competition and ruderalism (CSR) and the stress hypothesis used here. Maybe the authors could point out important differences. Or argue why they don’t use Grime’s theory which seems perfect for dune systems.
• Structure: I think the results and discussion part is somewhat mixed – so is the discussion and conclusion part (see specific comments).
• Figures: There are a lot of figures! Maybe some of them can go in appendix – or be transformed to tables. Or the whole paper should be rewritten to only focus on the initial hypotheses stated in the introduction – this would decrease the number of figures. See specific comments.

Experimental design

• Research question: Research questions are well defined in the introduction but a lot of add-ons are introduced later on (in the methods section). In general I think the paper could benefit from limiting the number of experiments and focusing on the most relevant ones.

Validity of the findings

• Conclusion: consider a more concise conclusion (see specific comments).

Additional comments

Specific comments
Introduction
Line 54-56: Sand drift and sand burial are equally harsh (if not harsher) to dune plants (as salty conditions) – I think it should be mentioned here also as it highly affect the species composition along the dune-inland gradient!
Line 73: I am not sure I like the ‘ecosystem service’ angle here. Is providing habitat for (threatened) species an ecosystem service? I think not – if ecosystem service is defined as something humans profit on (like pollination of crops)? Species diversity has its own right and it is not necessary to ‘justify’ by lumping it into ‘ecosystem services’.
Line 74: I think it is a bit of a statement to say that most research on sand dune communities is outdated and descriptive! A lot of recent work is quite the opposite… maybe rephrase to ‘historically, most research…’ or specify that research within your field (‘foundation species’, stress hypothesis) is missing - or something similar (for literature examples see studies by e.g. S. Provoost, M.A. Maun, A.K. Brunbjerg, E. Forey,).
Line 118-131: I am not sure I can follow the reasoning behind the term ‘elevation gradient’ and the changes of important processes along it. To me, elevation can be misunderstood – most often you will see the gradient in dunes described as a sea-inland gradient along which stress and disturbance decrease.
Also, if using the sea-inland gradient stress does not necessarily decrease inland as many dunes (at least in Europe) ‘end’ in dwarf shrub dominated dune heaths – these are very stressful environments because of the low pH! Maybe you need a more precise definition of stress versus disturbance here? I would say you work along a disturbance gradient – not a stress gradient (sensu Grime)!
Line 125: maybe you could point out that ‘foundation species’ may actually just be the sand-binders found mainly in fore dunes?
Materials and methods
Line 151: should Figure 2 be called figure 1 instead – as this is the first figure you refer to in the manuscript?
Line 160: how can you ‘estimate plant species’? Do you mean plant species frequency?
Line 161: replace sampled/zone with sampled in each zone or sampled per zone.
Line 174: this is the first time you mention Armeria – please give the full name!
Line 164-174: this is kind of an add-on to the original hypotheses you wanted to test (and stated at the end of the introduction) – is this important? If it is, I think it should be included in the study questions/hypotheses in the introduction – otherwise it is just a confusing add- on.
Line 178: replace 3 with ‘three’
Line 185: same as above
Line 201-213: again, an add-on to the original hypotheses. I suggest, you at least state your hypothesis related to the observed phenomenon here. What is the point of measuring temperature? What do you expect to find?
Line 218-220: the species pairs you choose are presumably the most dominant species – right? Maybe you can give an estimate on how dominant they are as a justification?
Line 230: state why you transform the Carpobrotus data.
Line 235-249: again, an add-on… see comment to line 201-213. I only see a reason to include these add-ons if you think they may affect the results of the original hypotheses.
Line 261: transformed how?
Line 263-326: How are these experiments/data related to the three original hypotheses/study questions at the end of the introduction? Please clarify in the text (either here (e.g. by adding a more specific section header) or in the introduction/study questions).
Line 289: do you mean seedling growth?
Line 293: replace 3 with ‘three’
Line 302: same as above
Results
Line 413-343: Results and discussion seem a bit mixed here: e.g you conclude on what causes the changes in species cover (‘competitively depressing’, ‘facilitating effect on’)… just keep it to the change in cover here – and the state the possible causes in the discussion.
Line 440-443: This belongs in the discussion.
Line 455-457: This belongs in the discussion
Line 511: replace different by differ
Discussion + conclusion: To me, much of what is written in the conclusion part belongs in the discussion. Keep the conclusion part concise and punchline like. Conclusion could maybe be line 656-684 or even only line 673-684.
Line 609: Replace 3 with three

Figures: I would like to see a figure showing a map of Sardinia + Italy and the location of your plot. 12 figures is a lot! Maybe some of them could go in appendix or be transformed to tables?
Fig 1: Do we need the full name of the plants and not just the genus name here?
Fig 3: Maybe add to the figure text how many years the experiment was running.
Fig 4: Do we need the full name of the plants and not just the genus name here (in the figure text)?

---

## Round 0.2 · accepted · Accept

Many thanks for your thorough response to my and the reviewers' comments.

Reviewer 2 ·

Basic reporting

I appreciate the revision and have no further comments to the manuscript

Experimental design

I have no further comments to the manuscript

Validity of the findings

I have no further comments to the manuscript

Additional comments

I appreciate the revision and have no further comments to the manuscript